# The Impact of Myosteatosis Percentage on Short-Term Mortality in Patients with Septic Shock

**DOI:** 10.3390/jcm11113031

**Published:** 2022-05-27

**Authors:** June-sung Kim, Jiyeon Ha, Youn-Jung Kim, Yousun Ko, Taeyong Park, Kyung Won Kim, Won Young Kim

**Affiliations:** 1Department of Emergency Medicine, University of Ulsan College of Medicine, Asan Medical Center, Seoul 05505, Korea; jsmeet09@gmail.com (J.-s.K.); yjkim.em@gmail.com (Y.-J.K.); 2Department of Radiology, Hallym University College of Medicine, Kangdong Seong-Sim Hospital, Seoul 05355, Korea; jiyeon.ha85@gmail.com; 3Biomedical Research Center, Asan Institute for Life Science, Asan Medical Center, Seoul 05505, Korea; ko.yousun82@gmail.com; 4Department of Radiology and Research Institute of Radiology, University of Ulsan College of Medicine, Asan Medical Center, Seoul 05505, Korea; pak14kr@naver.com (T.P.); medimash@gmail.com (K.W.K.)

**Keywords:** myosteatosis, mortality, septic shock, sepsis

## Abstract

The impact of myosteatosis on septic patients has not been fully revealed. The aim of the study was to evaluate the impact of the myosteatosis area and percentage on the 28-day mortality in patients with septic shock. We conducted a single center, retrospective study from a prospectively collected registry of adult patients with septic shock who presented to the emergency department and performed abdominal computed tomography (CT) from May 2016 to May 2020. The myosteatosis area defined as the sum of low attenuation muscle area and intramuscular adipose tissue at the level of the third lumbar vertebra was measured by CT. Myosteatosis percentages were calculated by dividing the myosteatosis area by the total abdominal muscle area. Of the 896 patients, 28-day mortality was 16.3%, and the abnormal myosteatosis area was commonly detected (81.7%). Among variables of body compositions, non-survivors had relatively lower normal attenuation muscle area, higher low attenuation muscle area, and higher myosteatosis area and percentage than that of survivors. Trends of myosteatosis according to age group were different between the male and female groups. In subgroup analysis with male patients, the multivariate model showed that the myosteatosis percentage (adjusted OR 1.02 [95% CI 1.01–1.03]) was an independent risk factor for 28-day mortality. However, this association was not evident in the female group. Myosteatosis was common and high myosteatosis percentage was associated with short-term mortality in patients with septic shock. Our results implied that abnormal fatty disposition in muscle could impact on increased mortality, and this effect was more prominent in male patients.

## 1. Introduction

Sepsis and septic shock are still leading causes of in-hospital death in worldwide [1]. Because of its heterogeneity, identifying which patients are at high risk of poor outcomes is needed for personalized management. The Third International Consensus Definitions for Sepsis and Septic Shock proposed to use the Sequential Organ Failure Assessment (SOFA) score for screening and prognostication [2]. However, assessing only organ dysfunction could not predict poor outcomes effectively, and intensive research efforts had tried to find out other risk factors for septic shock. In this context, focus was placed on body composition as a potential biomarker to predict clinical outcomes [3,4,5]. The body composition measurement could be carried out using computed tomography (CT) performed as a routine clinical practice [6,7,8,9]. To find out the source of infection and pre-intervention workup, abdominal CT evaluation in patients with septic shock is prevalent.

We previously reported that a decreased skeletal muscle index was related with poor outcomes in cancer patients with septic shock [10]. However, the skeletal muscle index is limited in that it only reflects muscle quantity, not function or strength [10]. Recent studies demonstrated that muscle quality and strength are associated with the degree of fatty infiltration or fatty degeneration called myosteatosis [11,12]. Myosteatosis, which once referred to age-related excessive fatty disposition, is now considered to be a pathological phenomenon and can be estimated by the attenuation of skeletal muscle Hounsfield Units (HU) on a CT scan [13,14]. Although impact of myosteatosis in patients with malignancy, liver cirrhosis, and inflammatory bowel disease had been reported, little is known about the impact of myosteatosis on the outcome of the patients with septic shock [15,16,17,18]. Recent studies with critically ill patients proved that abnormal fat disposition could be related with poor prognosis due to an overactivated inflammatory response [19,20,21]. However, their sample sizes were relatively small and detected myosteatosis with different definitions. Furthermore, mean muscle attenuation for the entire muscle area (MA) could be inaccurate for detecting the exact amount of myosteatosis because the proportion of myosteatosis might be markedly dependent on the entire MA [22]. Therefore, we tried to evaluate the prevalence of abnormal myosteatosis percentages, defined as myosteatosis area normalized by total abdominal MA, and the association between myosteatosis percentage and short-term mortality in patients with septic shock.

The objective of this study was to determine the prognostic significance of myosteatosis percentage in patients with septic shock.

## 2. Materials and Methods

### 2.1. Study Design and Population

This single center, observational, prospectively collected registry-based study was performed at the emergency department (ED) of a tertiary, university-affiliated hospital in Seoul, Korea, with an annual census of approximately 120,000 visits between May 2016 and May 2020. ED of the study facility have enrolled all adult (≥18 years old) patients with suspected or confirmed septic shock in the registry to monitor and improve outcomes [23]. Shock due to suspected infection was screened and enrolled to the registry by the emergency physicians on duty. We used a definition of septic shock as refractory low blood pressure (mean arterial pressure ≤ 65 mmHg) requiring vasopressors in spite of sufficient volume infusion or a serum lactate level ≥ 4 mmol/L, based on previous definition [24,25]. Patients were excluded in the registry if they were transferred from other hospital after initial resuscitation, were transferred to other hospital, had “do-not-resuscitation” order, refused to manage, developed septic shock 6 or more hours after ED arrival, or do not required vasopressor after fluid loading. All enrolled patients were equally treated with protocol-driven resuscitation following the current guidelines and bundles of the Surviving Sepsis Campaign [24].

This study included registry-enrolled patients who underwent abdominal CT examination, which had been taken for diagnostic purposes, at ED presentation. The institutional review board of the study facility approved this study (IRB number: 2021-0392) and waived the requirement for informed consent because of the retrospective characteristics.

### 2.2. Measures

Demographic data, focus of infection, SOFA score, acute physiology and chronic health evaluation (APACHE) II score, and initial lactate level were extracted from the registry. SOFA and APACHE II scores were calculated using the worst variables during the initial 24 h after ED presentation. The primary outcome was 28-day mortality.

Electronic medical records were used to collect additional data to assess body morphometry, such as body weight, height, and the presence of abdominal CT scan. Body mass index (BMI) was computed as the weight in kilograms divided by the height squared in meters (kg/m^2^). The body composition and quality, including subcutaneous fat area (SFA), visceral fat area (VFA), skeletal muscle area (SMA), normal attenuation muscle area (MA), intra-muscular adipose tissue area, and low attenuation MA, were assessed at the third lumbar (L3) vertebral level of abdominal CT scan performed at ED presentation using a previously developed web-based iAID toolkit [26]. Briefly, this algorithm selected a single axial image at the inferior endplate level of the L3 vertebra at the portal venous phase and segmented abdominal body compartments into three areas (i.e., total abdominal MA, SFA, and VFA). Moreover, total abdominal MA was divided into 3 muscle components using the pixel-wise measurement of CT density with the following range of HU for each component: normal attenuation MA with +30 HU to +150 HU, low attenuation MA with =29 HU to +29 HU, and intra-muscular adipose tissue area with −190 HU to −30 HU [13,27]. The accuracy of the generated muscle quality map composed of normal attenuation MA, low attenuation MA, and intra-muscular adipose tissue area was confirmed by a board-certified abdominal radiologist by comparing the original CT image and the muscle quality map image (Figure 1).

The skeletal muscle index (SMI) was calculated as the SMA which was defined as the sum of normal attenuation MA and low attenuation MA in cm^2^ divided by the height squared in meters (cm^2^/m^2^) [28]. Sarcopenia, diagnosed based on the abnormal SMI, was diagnosed based on the low muscle mass and impaired function, defined using sex-specific SMIs: <43 cm^2^/m^2^ for a BMI <25 kg/m^2^, <53 cm^2^/m^2^ for a BMI or 25 kg/m^2^ or more, and <41 cm^2^/m^2^ regardless of the BMI for females in a past study [28,29]. Myosteatosis was defined as the sum of low attenuation MA and intra-muscular adipose tissue area. Furthermore, the myosteatosis percentage was calculated as the myosteatosis area divided by the total abdominal MA to adjust the effect of total muscle amount. Abnormal myosteatosis area according to both age and sex was defined as in a previous report with healthy Asian populations [26].

### 2.3. Statistical Analyses

Descriptive statistics were stratified by 28-day all-cause mortality (i.e., survivor and non-survivor). Baseline demographics, clinical characteristics, laboratory results, and variables relating to body composition are presented as the frequency and percentage for categorical and median with interquartile range (IQR) for continuous variables. The Kolmogorov–Smirnov test was used to find out the normality of the distribution. Categorical variables were analyzed using Chi-squared or Fisher’s exact tests. We also evaluated the association between the skeletal muscle index, myosteatosis percentage and 28-day mortality separately according to gender. Univariate logistic regression tests were conducted with potential risk factors, which showed differences between survivors and non-survivors. Multivariate logistic regression was conducted with the variables that had previously well-known risk factors, such as age, infection focuses, comorbidities, lactate levels, SOFA, and APACHE score, with significantly different (*p*-values < 0.1) in the univariate logistic regression analysis [1,2,23]. Multicollinearity among variables was checked by using variance inflation factors before performing the multiple regression model. Moreover, we conducted subgroup analysis according to each gender because the previous report presented that the distributions of most body compositions were different between genders (10). We considered *p*-values less than 0.05 as statistically significant. Analyses were performed using SPSS Statistics for Mac, version 26 (IBM Corp., Armonk, NY, USA) and R version 3.5.0 (R foundation for Statistical Computing, Vienna, Austria).

## 3. Results

Among 1160 patients with septic shock in the registry, 896 (77.2%) patients who underwent an abdominal CT at ED were included. The rate of 28-day mortality was 16.3% (n = 146) (Figure 2).

### 3.1. Baseline Characteristics of the Cohort Population

The baseline characteristics of the study population are presented in Table 1. The median age was 67.0 (58.0–75.0) years with male predominance (58.8%). Non-survivors had higher proportions of chronic pulmonary disease (8.9 vs. 4.5%), malignancy (50.7 vs. 40.4%), hematologic disorder (11.6 vs. 5.5%), and liver cirrhosis (21.2 vs. 14.4%) than that of survivors. Except for unknown focus and blood stream, most sites of infection were significantly different between groups. Non-survivors showed more frequent infection in lung (24.0 vs. 14.9%) and intra-abdominal sources (22.6 vs. 14.4%) but less in urinary tract (9.6 vs. 16.9%) and hepato-biliary-pancreas (32.2 vs. 41.5%) than survivors. In addition, the non-survivor group had significantly higher lactate level, SOFA score, and APACHE scores than the survivor group. Based on diagnosing with current cut-off values, sarcopenia (63.3%) and myosteatosis (81.7%) were common in the study population, but there were no significant differences between the two groups. Separate demographics and clinical characteristics of each gender were showed in Appendix A.

### 3.2. Distribution of Myosteatosis Area and Percentage

Figure 3 illustrates the median of the myosteatosis area and percentage according to age groups for male and female patients. In the total population, the myosteatosis area had a positive correlation with age. Non-survivors had a larger area than survivors. These trends became clearer after converting myosteatosis percentages. The trends of the myosteatosis area and percentage according to age groups showed differences between the male and female groups. Differences between survivors and non-survivors were prominent in the forties to fifties in the female group and sixties to seventies in the male group.

The detailed body morphometry in the overall patients and by sex are presented in Table 2. Comparing with female, body compositions of male tent to show significant differences between survivor and non-survivor groups. Male non-survivors had smaller SMA (114.4 vs. 120.3 cm^2^), smaller normal attenuation MA (60.3 vs. 73.3 cm^2^), smaller total abdominal MA (128.8 vs. 138.4 cm^2^), larger low attenuation MA (53.8 vs. 49.0 cm^2^), and larger myosteatosis percentage (51.8 vs. 46.1%). However, these differences did not appear in female patients except for low attenuation MA (54.3 vs. 46.5 cm^2^) and myosteatosis percentage (69.1 vs. 60.9%).

### 3.3. Risk Factors of Short-Term Mortality

The multivariable logistic regression analyses for male patients showed that the lower SMI (adjusted OR 0.96; 95% CI 0.93–0.98; *p* = 0.03) and higher myosteatosis percentage (adjusted OR 1.03; 95% CI 1.01–1.05; *p* = 0.04) were independently associated with a higher 28-day mortality (Table 3). Meanwhile, myosteatosis percentages were not associated with 28-day mortality in female.

## 4. Discussion

In this study of body composition involving patients with septic shock, in which SMI was included as independent parameter, the myosteatosis percentage was the parameter that was associated with short-term survival. Our results implied that a high fatty disposition in muscle is associated with increased mortality in patients with septic shock and could be a useful prognostic factor in clinical practice. In addition, this effect was more prominent in male than female patients.

Myosteatosis is an abnormal fat depot that increases with aging and is identified to negatively correlate with muscle strength and mobility and impede metabolism [30]. Our result shows that myosteatosis is predominant in patients with septic shock. A recent review reported that the overall prevalence of myosteatosis in critically ill patients ranged from 25% to 40% [17,31]. This relatively low prevalence was attributed by employing a different definition of myosteatosis from that of the present study. Traditionally, myosteatosis was quantified by using the mean muscle density of the total abdominal muscle [22]. This definition could be inaccurate to detect the extent of myosteatosis and might under- or over-estimate the proportion of the ectopic fat deposit. Moreover, most studies set single cut-off values (e.g., below 40 HU) for detecting myosteatosis [22]. However, previous research with healthy population showed that the degree of myosteatosis was totally different according to age and sex. In this manner, we measured the direct area of normal attenuation MA and intra-muscular adipose tissue and diagnosed myosteatosis based on previously reported cut-off values according to age and sex [26]. Interestingly, this method revealed that a surprising portion of patients with septic shock had myosteatosis and the differences were largely due more to normal attenuation MA than to intra-muscular adipose tissue in both male and female groups.

We identified the statistical differences of myosteatosis percentages between 28-day survivors and non-survivors for both male and female patients. This scaling has a definite physiological basis that the myosteatosis area was largely affected by total abdominal MA. Although causality can be inferred only from direct controlled trials, we found that myosteatosis could be a surrogate marker of short-term survival among patients with septic shock. This result is consistent with that of a previous study that the progression of myosteatosis contributed to the development of sepsis [32]. Furthermore, a recent longitudinal laboratory study with mice reported that the amount of myosteatosis had a positive correlation with the severity of inflammation in non-alcoholic fatty liver disease [33]. The exact pathologic mechanism has not been completely revealed, but it seems to be the result of a complex cascade involving dysregulation of inflammatory response by excess adipose tissue [34]. A recent pre-clinic study also showed that worse muscle quality was related with abnormally higher levels of expression of inflammatory markers, suggesting the existence of abnormal response to pathogen [35].

Although the myosteatosis percentages were different between survivors and non-survivors, only the male group showed that this variable was an independent risk factor for predicting 28-day mortality. Sex-specific differences have been largely reported for fat distribution, properties, secretory function, and fatty acid handling in healthy participants and preclinical models [36]. The exact mechanism of this dimorphism is not yet elucidated; however, a plausible explanation is that there is a different proportion of sex hormones because sex steroids seem to play as endogenous modulators of development and function, and also influence the distribution of fat [37,38]. Another possible reason was that the study population consisted of an older age group (i.e., above 70% of patients were above sixty) so that differences in myosteatosis percentage were less evident between survivors and non-survivors in the female group. This proportion of age group might affect the results of the study. A further study with a larger sample size could reveal differences between gender groups.

Our study has several limitations. First, because the single-center retrospective design is in South Korea, it might be hard to generalize to other populations. Racial-ethnic differences in body composition have been widely reported [39]. Because there were no definite cut-off values of myosteatosis area or percentage, we diagnosed myosteatosis based on a previous report and this could over- or under-estimate the prevalence [26]. Second, we analyzed the patients with septic shock who clinically recorded CT images. Therefore, baseline characteristics of the study population showed a relatively low proportion of lung infection and a high proportion of abdominal infection. The relatively low mortality (16.5%) in the study was attributed to this selection bias. A further, well-designed, prospective study would be necessary to overcome this limitation. Third, although we tried to adjust the confounding factors, hidden confounders, including supplemental nutrients during the treated period, could affect mortality.

In conclusion, most patients with septic shock had myosteatosis, which was independently associated with 28-day mortality. Even though this association was evident in the male group, myosteatosis could be a useful prognostic marker for mortality in patients with septic shock.

## Figures and Tables

**Figure 1 jcm-11-03031-f001:**
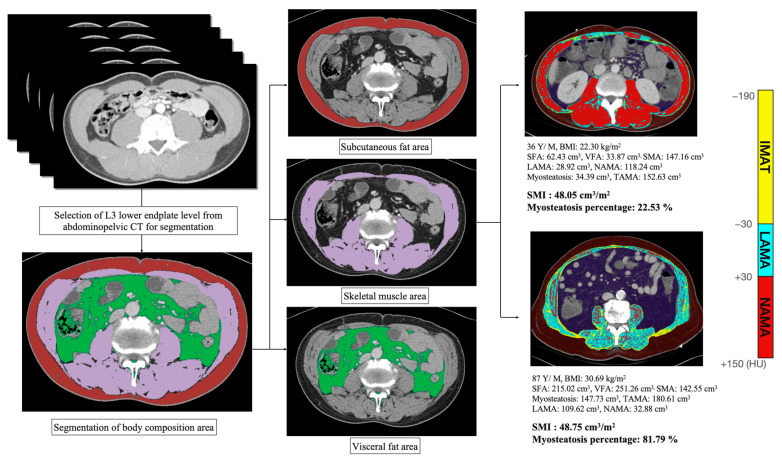
Example of body composition measurements. Muscle quality map generation using a web-based toolkit. After dividing body composition into 3 parts (i.e., subcutaneous fat, visceral fat, and skeletal muscle), skeletal muscle was classified in 3 areas according to certain attenuations (i.e., NAMA, LAMA, and IMAT). Myosteatosis area was defined as the sum of LAMA and IMAT. IMAT, inter/intramuscular adipose tissue area; LAMA, low-attenuation muscle area; NAMA, normal-attenuation muscle area; SMA, skeletal muscle area; TAMA, total abdominal muscle area.

**Figure 2 jcm-11-03031-f002:**
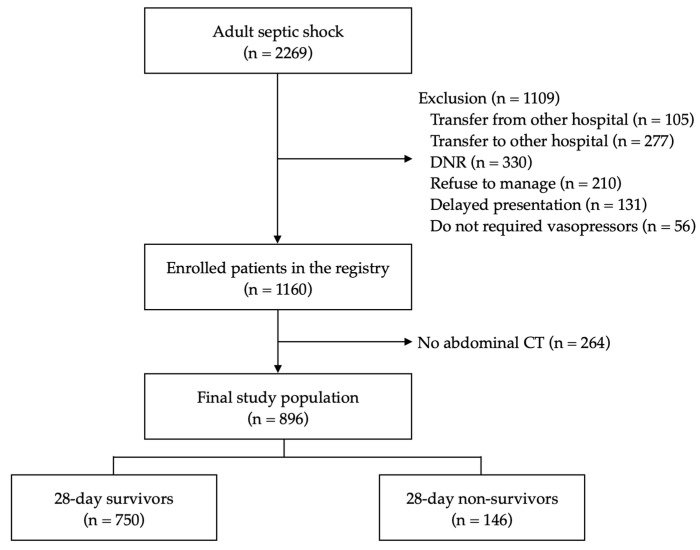
Flowchart of patient enrollment and allocation in the study. DNR, do-not-resuscitation; CT, computed tomography.

**Figure 3 jcm-11-03031-f003:**
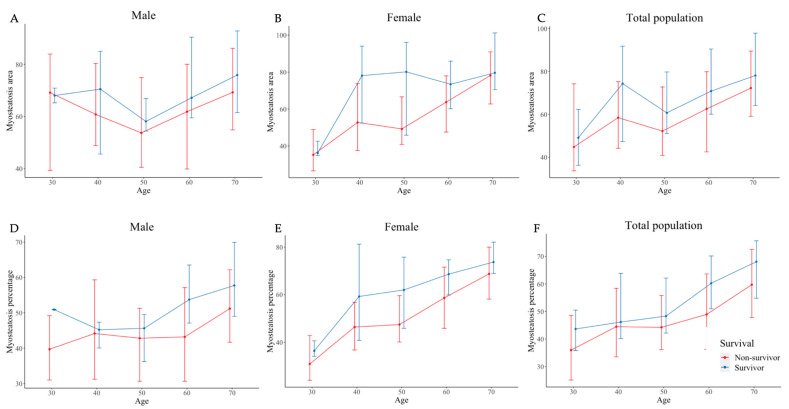
Distribution of myosteatosis area (**A**–**C**) and percentage (**D**–**F**) according to the age groups in male, female, and total patients.

**Table 1 jcm-11-03031-t001:** Baseline characteristics of the study population.

Characteristics	Total (n = 896)	Survivor (n = 750)	Non-Survivor (n = 146)	*p*-Value
Age	67.0 (58.0–75.0)	69.0 (60.0–76.0)	63.5 (53.8–69.0)	0.11
Male	527 (58.8)	433 (57.7)	94 (64.4)	0.14
Past illness				
HTN	299 (33.4)	253 (33.7)	46 (31.5)	0.60
DM	230 (25.7)	195 (26.0)	35 (24.0)	0.61
CAD	77 (8.6)	69 (9.2)	8 (5.5)	0.14
Chronic pulmonary disease	47 (5.2)	34 (4.5)	13 (8.9)	0.03
Malignancy	377 (42.1)	303 (40.4)	74 (50.7)	0.02
Hematologic disorder	58 (6.5)	41 (5.5)	17 (11.6)	<0.01
CKD	51 (5.7)	45 (6.0)	6 (4.1)	0.37
LC	139 (15.5)	108 (14.4)	31 (21.2)	0.04
Site of infection				
Unknown	82 (9.2)	67 (8.9)	15 (10.3)	0.61
Lung	147 (16.4)	112 (14.9)	35 (24.0)	<0.01
Urinary tract	141 (15.7)	127 (16.9)	14 (9.6)	0.03
Intra-abdomen	141 (15.7)	108 (14.4)	33 (22.6)	0.01
Hepato-biliary-pancreas	358 (40.0)	311 (41.5)	47 (32.2)	0.04
Blood stream	66 (7.4)	51 (6.8)	15 (10.3)	0.20
Lactate level	3.6 (1.9–5.8)	3.2 (1.7–5.4)	5.4 (2.6–9.2)	<0.01
SOFA score	7.0 (5.0–10.0)	7.0 (5.0–9.0)	10.0 (6.0–13.0)	<0.01
APACHE score	15.0 (11.0–20.0)	13.0 (11.0–23.0)	17.0 (11.0–23.0)	<0.01
Sarcopenia	567 (63.3)	474 (63.2)	93 (63.7)	0.91
Myosteatosis	732 (81.7)	607 (80.9)	125 (85.6)	0.19

Data are presented as n (%) or median (interquartile range). Abbreviations: HTN, hypertension; DM, diabetes mellitus; CAD, coronary artery disease; CKD, chronic kidney disease; LC, liver cirrhosis; SOFA, sequential organ failure assessment; APACHE, acute physiology and chronic health evaluation.

**Table 2 jcm-11-03031-t002:** Body composition of the study population.

Body Composition	Total (n = 896)	Survivor (n = 750)	Non-Survivor (n = 146)	*p*-Value
**Total**				
BMI, kg/m^2^	22.2 (19.7–24.6)	22.3 (19.7–24.6)	21.9 (19.7–24.9)	0.64
SFA, cm^2^	107.3 (65.7–157.0)	110.0 (66.6–157.2)	94.9 (56.6–150.5)	0.07
VFA, cm^2^	101.0 (57.7–158.9)	102.3 (57.4–160.4)	90.9 (59.0–147.7)	0.20
SMA, cm^2^	106.2 (90.6–125.6)	106.1 (89.8–126.7)	106.7 (93.8–119.6)	0.76
^1^ SMI, cm^2^/m^2^	40.6 (36.3–46.0)	40.6 (36.3–46.4)	41.0 (36.0–44.6)	0.67
Normal attenuation MA, cm^2^	56.8 (37.8–78.4)	57.4 (39.6–79.7)	52.8 (32.8–69.6)	<0.01
Intramuscular adipose tissue area, cm^2^	16.0 (9.6–23.2)	16.3 (9.7–23.3)	14.9 (9.2–22.6)	0.42
Low attenuation MA, cm^2^	49.2 (36.8–60.9)	47.6 (36.0–59.5)	54.1 (42.7–66.4)	<0.01
^1^ Total abdominal MA, cm^2^	123.5 (109.2–142.1)	123.2 (109.0–143.3)	124.9 (109.9–136.2)	0.59
^2^ Myosteatosis area, cm^2^	66.6 (47.5–84.1)	65.9 (46.6–83.3)	70.7 (55.2–90.5)	0.03
^3^ Myosteatosis percentage, %	0.53 (0.40–0.67)	0.53 (0.39–0.66)	0.57 (0.44–0.71)	<0.01
**Male**				
BMI, kg/m^2^	22.0 (19.7–24.4)	22.1 (19.7–24.4)	21.6 (20.0–24.4)	0.63
SFA, cm^2^	87.7 (52.6–131.1)	88.8 (55.8–131.3)	83.9 (49.6–123.8)	0.26
VFA, cm^2^	110.1 (60.4–170.2)	113.1 (59.0–171.7)	97.2 (65.4–158.0)	0.30
SMA, cm^2^	118.9 (105.3–137.2)	120.3 (106.2–138.3)	114.4 (101.0–125.9)	<0.01
^1^ SMI, cm^2^/m^2^	42.7 (38.5–49.3)	42.8 (38.6–49.6)	42.1 (37.4–46.0)	0.07
Normal attenuation MA, cm^2^	70.2 (52.0–89.3)	73.3 (54.3–92.2)	60.3 (45.5–73.0)	<0.01
Intramuscular adipose tissue area, cm^2^	14.9 (8.6–21.1)	14.9 (8.7–21.1)	14.0 (7.5–21.7)	0.57
Low attenuation MA, cm^2^	50.0 (36.6–62.8)	49.0 (35.3–62.6)	53.8 (44.8–65.0)	0.02
^1^ Total abdominal MA, cm^2^	135.2 (120.4–152.6)	138.4 (121.3–154.6)	128.8 (115.1–144.7)	<0.01
^2^ Myosteatosis area, cm^2^	65.9 (46.3–84.1)	65.5 (45.7–83.5)	67.0 (55.2–86.4)	0.13
^3^ Myosteatosis percentage, %	47.7 (35.1–60.2)	46.1 (33.7–59.4)	51.8 (40.5–66.2)	<0.01
**Female**				
BMI, kg/m^2^	22.6 (19.7–24.9)	22.6 (19.7–24.8)	22.4 (19.6–25.8)	0.96
SFA, cm^2^	137.8 (93.8–181.4)	138.4 (95.6–181.4)	124.9 (81.0–186.6)	0.53
VFA, cm^2^	91.1 (53.8–133.0)	91.7 (55.6–136.4)	85.4 (33.3–130.1)	0.27
SMA, cm^2^	90.6 (80.0–100.8)	90.9 (80.3–100.4)	89.3 (79.1–104.0)	0.51
^1^ SMI, cm^2^/m^2^	37.9 (34.1–41.7)	37.8 (33.9–41.5)	37.9 (34.1–43.6)	0.60
Normal attenuation AMA, cm^2^	41.0 (28.2–55.9)	42.4 (28.6–56.0)	33.7 (23.5–53.3)	0.08
Intramuscular adipose tissue area, cm^2^	17.9 (12.0–25.7)	18.0 (11.9–25.5)	16.8 (12.6–27.0)	0.71
Low attenuation MA, cm^2^	48.0 (36.9–58.7)	46.5 (36.6–57.8)	54.3 (40.6–66.6)	<0.01
^1^ Total abdominal MA, cm^2^	109.8 (98.8–121.7)	109.3 (98.7–120.9)	114.8 (99.1–127.7)	0.29
^2^ Myosteatosis area, cm^2^	67.3 (49.4–84.3)	66.8 (48.7–83.0)	75.2 (54.0–95.2)	0.13
^3^ Myosteatosis percentage, %	61.5 (47.8–75.1)	60.9 (47.3–74.6)	69.1 (53.1–77.7)	0.05

Data are presented as median (interquartile range). ^1^ Total abdominal MA was derived by adding the Normal attenuation MA, Intramuscular adipose tissue area, and Low attenuation MA. ^2^ Myosteatosis area was derived by adding the Low attenuation MA and Intramuscular adipose tissue area. ^3^ Myosteatosis proportion was defined as myosteatosis divided by Total abdominal MA. Abbreviations: BMI, body mass index; SFA, subcutaneous fat area; VFA, visceral fat area; SMA, skeletal muscle area; SMI, skeletal muscle area index; MA, muscle area.

**Table 3 jcm-11-03031-t003:** Multivariate analysis of septic shock patients for its association with 28-day mortality.

Variables	Univariate Analysis			Multivariate Analysis		
	OR	95% CI	*p*-Value	Adjusted OR	95% CI	*p*-Value
**Male**						
Age	0.99	0.97–1.01	0.49			
Intra-abdominal infection	1.63	0.86–3.11	0.13	1.92	1.06–3.47	0.03
Hepato-biliary-pancreas infection	0.71	0.41–1.24	0.71			
Blood stream infection	2.58	0.93–7.16	0.07	2.72	0.99–7.49	0.06
Lactate	1.20	1.11–1.29	<0.01	1.19	1.11–1.28	<0.01
SOFA	1.11	1.02–1.20	0.01	1.13	1.06–1.21	<0.01
APACHE	1.01	0.97–1.05	0.77			
SMI	0.96	0.94–0.99	0.02	0.96	0.93–0.98	0.03
myosteatosis percentage	1.13	1.09–1.16	0.04	1.03	1.01–1.05	0.04
**Female**						
Age	0.99	0.96–1.02	0.60			
Malignancy	2.20	1.12–4.29	0.02	2.20	1.13–4.28	0.02
Lactate	1.11	1.01–1.22	0.03	1.11	1.01–1.22	0.04
SOFA	1.27	1.16–1.40	<0.01	1.29	1.17–1.41	<0.01
Myosteatosis percentage	1.06	0.94–1.16	0.81			

Abbreviations: OR, odds ratio; CI, confidence interval; CAD, coronary artery disease; CKD, chronic kidney disease; SOFA, sequential organ failure assessment; APACHE, acute physiology and chronic health evaluation.

## Data Availability

Data sharing is not applicable to this article.

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
