# Peer review of "The Impact of Myosteatosis Percentage on Short-Term Mortality in Patients with Septic Shock"

_jcm, 2022, doi:10.3390/jcm11113031_

Round 1
Reviewer 1 Report
It is suggested that the authors include in the article how myoesteatosis can affect the clinical evolution and the outcome of septic shock patient, considering the dimension of the adipose compartment in relation to the total adipose deposit of the organism;
The document shows some formatting errors, as an example:
- reference “22” on line 231;
- reference “26” on line 238;
- words “among between” on line 242
Author Response
Comments and Suggestions for Authors
It is suggested that the authors include in the article how myoesteatosis can affect the clinical evolution and the outcome of septic shock patient, considering the dimension of the adipose compartment in relation to the total adipose deposit of the organism;
The document shows some formatting errors, as an example:
- reference “22” on line 231;
Response> Thanks for the generous comments. We corrected typos.
Aleixo, G.F.P.; Shachar, S.S.; Nyrop, K.A.; Muss, H.B.; Castillo, L.M.; Williams, G.R. Myosteatosis and prognosis in cancer: systematic review and meta-analysis. Crit Rev Oncol Hemat 2020, 145, 102839. (reference 22)
- reference “26” on line 238;
Response> We corrected the format.
Kim, D.W.; Kim, K.W.; Ko, Y.; Park, T.; Khang, S.; Jeong, H.; Koo, K.; Lee, J.; Kim, H.-K.; Ha, J.; Sung, Y.S.; Shin, Y. Assessment of myosteatosis on computed tomography by automatic generation of a muscle quality map using a web-based toolkit: feasibility study. Jmir Med Inform 2020, 8, e23049. (reference 26)
- words “among between” on line 242
Response> We corrected a typo.
“We identified the statistical differences of myosteatosis percentages between 28-day survivors and non-survivors for both male and female patients.” (line 242-243)
Reviewer 2 Report
First of all, thank you for giving me the opportunity to review this article.
It is a monocentric Korean retrospective study which aim was to evaluate the impact of myosteatosis area and percentage on the 28-day mortality in patients with septic shock.
This study is simple and well designed and brings a new message.
I have only minor comments to improve this manuscript.
First could you be more precise concerning the definition you use for septic shock. I don’t understand if you take into account lactate into your definition of septic shock. If not, is it possible to make a sub group analysis among the patients with lactate > 2 mmol/L ( or 4 mmol/L).
Then into the statistical analyses paragraph,
Did you check the log linearity of continuous variables tested in the logistic regression models?
Could you precise the cut off you choose in univariate analysis to test the variables into the multivariate analyses.
In the discussion, could you underline that your death rate is quite low (only 16.5% instead of the usual 40% of death rate in septic shock). Might be related to your septic shock definition.
The figure 3 could be more informative with interquartiles or confidence intervals. Then, it would be interesting to have a pvalue on the figure (ANOVA or equivalent) to know if the differences between subgroups are or not significant.
Author Response
Comments and Suggestions for Authors
First of all, thank you for giving me the opportunity to review this article.
It is a monocentric Korean retrospective study which aim was to evaluate the impact of myosteatosis area and percentage on the 28-day mortality in patients with septic shock.
This study is simple and well designed and brings a new message.
I have only minor comments to improve this manuscript.
First could you be more precise concerning the definition you use for septic shock. I don’t understand if you take into account lactate into your definition of septic shock. If not, is it possible to make a sub group analysis among the patients with lactate > 2 mmol/L ( or 4 mmol/L).
Response> Thanks for the generous comments. Because of the study period, the septic shock registry of the study facility was defined based on previous definition of sepsis (i.e., Sepsis-2) instead of recent definition (i.e., Sepsis-3). We added the sentences to clarify the definition of septic shock for this study.
“Shock due to suspected infection was screened and enrolled to the registry by the emergency physicians on duty. We used a definition of septic shock as refractory low blood pressure (mean arterial pressure ≤ 65 mmHg) requiring vasopressors in spite of sufficient volume infusion or a serum lactate level ≥ 4 mmol/L, based on previous definition [24, 25].” (line 77-80)
Then into the statistical analyses paragraph,
Did you check the log linearity of continuous variables tested in the logistic regression models?
Response> We agreed the reviewer’s comments that checking the continuous variables with log linearity tests before the logistic regression models. We performed the tests and there were no collinearities among variables. We added sentences as below.
“Multicollinearity among variables was checked by using variance inflation factors before performing the multiple regression model.” (line 147)
Could you precise the cut off you choose in univariate analysis to test the variables into the multivariate analyses.
Response> We used the cut-off value (< 0.1) in multivariate analyses and added in the manuscript.
“Multivariate logistic regression was conducted with the variables that had previously well-known risk factors, such as age, infection focuses, comorbidities, lactate levels, SOFA, and APACHE score, with significantly different (p-values < 0.1) in the univariate logistic regression analysis [1,2,23].” (line 144-147)
In the discussion, could you underline that your death rate is quite low (only 16.5% instead of the usual 40% of death rate in septic shock). Might be related to your septic shock definition.
Response> We agreed the reviewer’s concern that there were differences of mortality between this study and previous reports. Short-term (i.e., 28-day) mortality of entire population of the study period was around 30-35%. However, as we already mentioned in the limitation section, we could only include the patients with septic shock who clinically performed CT because of the retrospective design. Therefore, it was inevitable to include large proportion of patients with suspected abdominal infection and malignancies and exclude some proportion of patients with suspected pneumonia or blood stream infection those who had relatively poorer prognosis than that of other sources of infection. This selection bias might be solved through well-designed prospective study in the future. We mentioned these details in the discussion section as below.
“Second, we analyzed the patients with septic shock those who clinically performed CT images. Therefore, baseline characteristics of the study population showed relatively low proportion of lung infection and high proportion of abdominal infection. Relatively low mortality (16.5 %) of the study was attributed to this selection bias. Further well-designed prospective study would be necessary to overcome this limitation.” (line 283-289)
The figure 3 could be more informative with interquartiles or confidence intervals. Then, it would be interesting to have a pvalue on the figure (ANOVA or equivalent) to know if the differences between subgroups are or not significant.
Response> We totally agreed the reviewer’s opinion that the figure 3 could be more informative with interquartile or confidence intervals. We added the median with interquartile range of myosteatosis area and percentage for each population in figure 3. Regretfully, we could not provide individual p-values of differences between survivors and non-survivals for specific age group because of not enough sample size.